# Profiling Substrate Specificity of the SUMO Protease Ulp1 by the YESS–PSSC System to Advance the Conserved Mechanism for Substrate Cleavage

**DOI:** 10.3390/ijms232012188

**Published:** 2022-10-13

**Authors:** Faying Zhang, Hui Zheng, Yufan Xian, Haoyue Song, Shengchen Wang, Yueli Yun, Li Yi, Guimin Zhang

**Affiliations:** 1School of Life Sciences, Hubei University, Wuhan 430062, China; 2College of Life Science and Technology, Beijing University of Chemical Technology, Beijing 100029, China

**Keywords:** SUMO, SUMO protease, substrate specificity, YESS–PSSC system

## Abstract

SUMO modification is a vital post-translational regulation process in eukaryotes, in which the SUMO protease is responsible for the maturation of the SUMO precursor and the deconjugation of the SUMO protein from modified proteins by accurately cleaving behind the C-terminal Gly–Gly motif. To promote the understanding of the high specificity of the SUMO protease against the SUMO protein as well as to clarify whether the conserved Gly–Gly motif is strictly required for the processing of the SUMO precursor, we systematically profiled the specificity of the *S. cerevisiae* SUMO protease (Ulp1) on Smt3 at the P2–P1↓P1’ (Gly–Gly↓Ala) position using the YESS–PSSC system. Our results demonstrated that Ulp1 was able to cleave Gly–Gly↓ motif-mutated substrates, indicating that the diglycine motif is not strictly required for Ulp1 cleavage. A structural-modeling analysis indicated that it is the special tapered active pocket of Ulp1 conferred the selectivity of small residues at the P1–P2 position of Smt3, such as Gly, Ala, Ser and Cys, and only which can smoothly deliver the scissile bond into the active site for cleavage. Meanwhile, the P1’ position Ala of Smt3 was found to play a vital role in maintaining Ulp1’s precise cleavage after the Gly–Gly motif and replacing Ala with Gly in this position could expand Ulp1 inclusivity against the P1 and P2 position residues of Smt3. All in all, our studies advanced the traditional knowledge of the SUMO protein, which may provide potential directions for the drug discovery of abnormal SUMOylation-related diseases.

## 1. Introduction

Post-translational modifications are an important mechanism for dynamic regulation of cellular proteins in eukaryotes. Among them, the SUMO modification system is a unique type, in which Small Ubiquitin-like Modifiers (SUMOs) are covalently conjugated to the lysine residue of a wide variety of target proteins to regulate their subcellular localization, activity, structure or stability [1,2,3]. The number of SUMO protein genes varies among different organisms. Lower organisms such as yeast, *Caenorhabditis elegans* and *Drosophila* contain a single SUMO gene in their genomes, whereas plants and vertebrates possess several copies [4,5,6,7,8]. The reported SUMO proteins are all expressed as immature precursors with C-terminal extensions of varying sequences (2–11 amino acids). Their maturation requires the removal of C-terminal extensions by the SUMO-specific protease to expose the diglycine motif (GG↓). Mature SUMO proteins are then conjugated stepwise to target proteins through the SUMO activating enzyme, the conjugating enzyme and ligase [9]. Upon finishing regulation, the SUMO modifier can be deconjugated from the target proteins by the SUMO-specific protease [9]. Obviously, the SUMO-specific protease, as an important regulator, maintains the balance of SUMO cycling by facilitating the maturation and deconjugation of the SUMO protein, whereas its deregulation would lead to cellular dysfunction and result in various diseases including cancer [10], heart disease [11] and neurodegenerative diseases [12]. Deciphering SUMO protease specificity benefits the advance in the understanding of the SUMO regulation system, providing potential directions for the drug discovery of abnormal SUMOylation-related diseases [13,14].

The *Saccharomyces cerevisiae* SUMO-specific protease Ulp1, as a representative model, was first studied to dissect the substrate specific mechanism, revealing that recognition of Ulp1 towards the SUMO protein Smt3 is achieved through an extensive interface involving numerous hydrophilic interactions and salt bridges and that the catalytic triad C580–H514–D531 enables the precise cleavage behind the conserved Gly–Gly motif in an extremely tight and shallow active tunnel [15]. The structural explanation for the highly-accurate cleavage behind the Gly–Gly↓ motif is that other residue substitutions at this position would possibly provoke a steric clash with the narrow active tunnel, thus affecting the cleavage efficiency. Despite being theoretically plausible, whether other motifs other than di-Gly can be effectively processed by Ulp1 remains unknown due to lack of in vivo or in vitro experimental results. In addition, a previous work reported that directly-expressed human SUMO-1/3–AA, –GS, –GN, –GA mutants could also SUMOylate intracellular proteins as efficiently as the wild-type SUMO protein [16]. This observation raised an interesting question of whether the Gly–Gly motif could be regarded as an essential hallmark of SUMO proteins. Besides the Gly–Gly (P2–P1) position, the P1’ position specificity (the first residue after the Gly–Gly motif) is also noteworthy because this position plays a crucial role in modulating the SUMO protease preference on divergent SUMO isoforms [17].

Compared to other proteases with an oligopeptide as a substrate, the property of cleaving protein substates makes the substrate specificity characterization of the SUMO protease more challenging [18,19]. The gel-electrophoresis-based in vitro method has previously been applied to profile Ulp1 specificity on the P1’ position residue of Smt3, in which Ulp1 and 20 protein substrates with arbitrary amino acids at the P1’ position were individually purified and reacted, followed by quantitation using SDS-PAGE and gel-scanning software [20]. Obviously, the diversity of substrates makes the mapping of one or more sites technically difficult and time-consuming. Several in vivo methods have been exploited to investigate the substrate specificity of peptide–substrate proteases in a high-throughput manner, such as the phage display system [21], the bacterial display system [22] and the yeast display system based analysis methods [18,23], whereas protein–substrate proteases, like the SUMO protease, have never been reported in these systems.

In this study, our newly developed Yeast ER sequestration screening-based protease–substrate specificity characterization (YESS–PSSC) system [18] was applied to systematically profile the substrate specificity of the *S. cerevisiae* SUMO protease (Ulp1) at the P2–P1↓P1’ (GG↓A) position of the SUMO protein (Smt3). Compared to conventional gel-electrophoresis-based in vitro assays, the YESS–PSSC system largely reflected the properties of the SUMO protease in physiological status and could easily quantify the diversity of substrates through a yeast surface display and flow cytometry technology. We experimentally verified that the recognition of Ulp1 towards Smt3 depends on an extensive interacting interface, so single residue mutations in Smt3 hardly affected Ulp1 cleavage efficiency. In addition, a systematical characterization of residue specificity against Smt3 at the P1 and P2 positions, followed by a structural modeling analysis revealed that the Gly–Gly motif is not strictly required for Ulp1 cleavage. Moreover, further exploration revealed the vital role of Ala at the P1’ position of Smt3 for Ulp1 substrate specificity.

## 2. Results and Discussion

### 2.1. Verification of the YESS–PSSC System for Quantitatively Characterizing Ulp1–Smt3 Specificity

Conventional gel-electrophoresis-based in vitro methods for studying the protease–substrate specificity are always complicated and time-consuming due to the large number of tested substrates [20]. In contrast, in vivo analytical platforms such as the YESS–PSSC system, which is used for studying the substrate specificity of Ulp1 against Smt3, are particularly attractive for their advantages on simplicity and high-throughput [18]. Based on the YESS–PSSC concept, Ulp1 and the bi-epitope tag fused substrate cassette (Aga2-FLAG-Smt3↓-HA) were simultaneously expressed under the control of a divergent promoter (*GAL*1-*GAL*10), followed by being escorted into yeast ER for proteolytic reactions. The proteolytic substrate part (Aga2-FLAG-Smt3↓) was then displayed on the cell surface for fluorophore-conjugated antibody labelling and flow cytometry analysis (Figure 1A).

The display efficiency of the substrate cassette determined the efficacy of the YESS–PSSC system. Considering that Smt3 is a globular-structure protein containing 101 residues [15], the display efficiency of the Smt3 substrate cassette (Aga2-FLAG-Smt3-HA) was evaluated first (Figure 1B). The Smt3 substrate cassette was displayed on the yeast cell surface with an efficiency of about 50–60%, which was slightly lower than the peptide–substrate display system’s efficiency of 70% [18]. However, the clearly-separated cell population and the utter dual anti-FLAG-iFluor 647/anti-HA-FITC fluorescence intensities implied that the Smt3 substrate cassette was well-expressed, cell-surface-displayed and not interfered by the *S. cerevisiae* endogenous Ulp1. This might be because the substrate part was escorted into yeast ER through the ER signal peptide, while the endogenous Ulp1 was usually localized in the nucleus, guided by its N-terminal nuclear-localization signal (NLS) [24]. Subsequently, we proceeded to evaluate Ulp1 activity towards the Smt3 substrate cassette in the YESS–PSSC system. *S. cerevisiae* Ulp1 contains an N-terminal regulation domain (1–402) and a highly conserved C-terminal protease domain (403–621), among which the C-terminal protease domain is sufficient to recognize and cleave Smt3 with comparably high activity to full-length Ulp1 [24]. The C-terminal protease domain was used in our studies to evaluate its substrate specificity. In principle, if Smt3 was recognized and cleaved by Ulp1, the C-terminal HA tag of the substrate cassette would be removed to generate the dominating display of Aga2-FLAG-Smt3GG↓ on the cell surface, resulting in high anti-FLAG-iFluor 647 and no–low anti-HA-FITC fluorescence intensities. However, if the substrate cassette was not cleaved, the intact Aga2-FLAG-Smt3-HA would be displayed and labeled with both high anti-FLAG-iFluor 647 and anti-HA-FITC fluorescence intensities. The stronger anti-HA-FITC fluorescence intensity indicated a more inefficient cleavage. As expected, only anti-FLAG-iFluor 647 fluorescence intensity was observed in the cells co-expressing Ulp1 and the Smt3 substrate cassette, suggesting nearly 100% cleavage efficiency (Figure 1B). These results collectively verified that the YESS–PSSC system is the proper system to study Ulp1 substrate specificity.

Not only can it be applied to characterize the substrate specificity of Ulp1, the YESS–PSSC system also showed great potential in dissecting the substrate specificity of other SUMO protease homologs as well as protein–substrate proteases, with several unique advantages. Firstly, the in vivo proteolytic reaction enabled the evaluation of multiple substrates without the need to purify the protease and substrates, so the process was time- and labor-efficient [18,19]. Secondly, compartmentalizing the proteolytic reaction of protease in the endoplasmic reticulum (ER) through the ER signal peptide largely maintained the physiological properties of proteases and substrates due to the existence of various molecular chaperones in the ER [25]. Thirdly, the powerful secretory system of *S. cerevisiae* enabled the proteolytic protein products to be quickly transported to the cell surface for fluorophore-conjugated antibody labelling and flow cytometry analysis [26,27]. Besides, the ER separated the proteolytic reaction from complex endogenous biological processes, thus barely being disturbed by the endogenous regulating system.

### 2.2. Validation of the YESS–PSSC System by Mapping Ulp1 Residue Specificity at the P1’ Position of Smt3

As the first residue after the substrate cleavage site as well as being extremely close to the catalytic triad, proteases normally show high specificity at the substrate P1’ position. As for Ulp1, its specificity for the P1’ position of Smt3 has previously been characterized through in vitro experiments, demonstrating that mutated Smt3 substrates with any residues at this position can be efficiently cleaved by Ulp1 except Pro [20]. In our work, this position was quantitatively characterized in vivo through the YESS–PSSC approach. Substrate cassettes (Aga2-FLAG-Smt3GG↓X-HA, where X is an arbitrary amino acid) were, respectively, co-expressed with Ulp1 and evaluated in the YESS–PSSC system, followed by flow cytometry analysis. The results showed that Ulp1 effectively cleaved 19 types of substrates with >95% activity except the Smt3–GG↓P substrate (Figure 2) which exhibited both high anti-FLAG-iFluor 647 and anti-HA-FITC fluorescence signals. Our results, similar to those of the gel-electrophoresis-based in vitro experiments, further verified the accuracy of the YESS–PSSC approach in studying the substrate specificity of the SUMO protease Ulp1 [20].

### 2.3. Single Mutation of Smt3 in the Smt3–Ulp1 Interacting Interface Would Not Affect Ulp1 Recognition Specificity

The crystal structure of the Ulp1–Smt3 complex revealed that the Ulp1–Smt3 interacting interface could be roughly described as six structural motifs of Ulp1 interacting with the exposed β sheet and C-terminal strand of Smt3 [15] (Figure 3A). Several conservative sites in the Ulp1 motif were, respectively, mutated to identify the essential roles of central regions such as motif 2, motif 3 and motif 5 in maintaining Ulp1–Smt3 recognition specificity [15]. In our work, the key residues of Smt3 interacting with Ulp1 in every motif were mutated to nonpolar amino acids to explore the significance of each site for Ulp1–Smt3 recognition specificity. Guided by the structural analysis, R64G, R68G, G69A, R71G, D82G, E94G, Q95G, I96G and G98A mutations were, respectively, introduced to eliminate the major salt bridge or/and hydrogen bond contact with Ulp1.

The mutated Smt3^I^ cassettes (Aga2-FLAG-Smt3^I^ GG↓A-HA, where Smt3^I^ represents the Smt3 interface mutation) were, respectively, co-expressed with Ulp1 in the YESS–PSSC system. Flow cytometry analysis showed that the wild-type Smt3 and eight mutated Smt3^I^ substrates were all cleaved by Ulp1 with more than 98% efficiency except the Smt3^I^–G98A mutant whose cleavage efficiency was 94.65%, indicating that a single mutation in Smt3 had almost no effect on its interactions with Ulp1 (Figure 3B). This can be explained by the recognition of Ulp1 towards Smt3 being dependent on an extensive interface consisting of multiple residues, thereby the influence caused by a single mutation could be neglected. It is worth noting that the Smt3–G98A mutation, a position that is usually considered highly-conserved in SUMO proteins (GG^98th^↓A), exhibited a cleavage efficiency that was about 5% lower than the wild-type Smt3 and other mutations. This confirmed that Gly98 is indeed important for the recognition and cleavage of Ulp1 against Smt3, and, more importantly, it also implied that the Gly–Gly motif is not strictly required for the Ulp1 cleavage of Smt3. Therefore, we further characterized the specificity of Ulp1 against the conserved Gly–Gly motif (P2-P1↓ position) of Smt3.

### 2.4. Characterizing Residue Specificity of Ulp1 against Smt3 at P1 and P2 Positions

To explore the residue specificity of Ulp1 against Smt3 at the P1 and P2 positions (the conserved Gly–Gly motif), a series of mutated Smt3^P^–GX_1_↓A and –X_2_G↓A substrates (where X is an arbitrary amino acid, and Smt3^P^ represents the Smt3 P1 or P2 mutation) were constructed and analyzed in the YESS–PSSC system. It needs to be emphasized that we mutated each Gly into larger amino acids because these two sites are both located in the narrow active tunnel of Ulp1 (Figure 4B) [15]. At the P1 position, Ulp1 presented a high activity of 94.65% against the Smt3^P^–GA↓A substrate, which is only slightly lower than the 99.48% cleavage efficiency against the wild-type Smt3–GG↓A substrate. Comparably, when replacing Gly at the P1 position with residues that have bulky side chains, the cleavage efficiency of Ulp1 against the Smt3^P1^ substrate gradually decreased, showing a cleavage efficiency of 46.07%, 29.04%, 8.60% and 2.59% for Smt3^P^–GS↓A, Smt3^P^–GC↓A, Smt3^P^–GT↓A and Smt3^P^–GW↓A substrates, respectively (Figure 4A). Other residue substitutions were not analyzed because it could be speculated that amino acids having bulkier side chains than Thr could possibly provoke steric hindrance, preventing Smt3^P^ from entering the substrate tunnel. At the P2 position, Ulp1 cleavage efficiency against the Smt3^P^–AG↓A, Smt3^P^–SG↓A and Smt3^P^–CG↓A substrates were, respectively, measured to 99.39%, 99.42% and 96.98% (Figure 4A). Similar to that of the P1 position, Ulp1 presented no detectable cleavage against Smt3 with Thr or Trp at the P2 position (Figure 4A). Based on these results, three dual-mutation substrates, including Smt3^P^–AA↓A, Smt3^P^–SA↓A and Smt3^P^–CA↓A, were further constructed to explore whether Ulp1 could accept dual mutations at the Gly–Gly motif. No detectable cleavage was observed on these substrates, even though each single-mutated substrate could be efficiently cleaved by Ulp1 with high efficiency (Figure 4A), implying the significance of the role of the smallest residue Gly at the P1 or P2 position for Ulp1 cleavage.

The C-terminal Gly–Gly motif is highly-conserved in all reported SUMO proteins from yeast to human, whereas our work suggested that Ulp1 was sufficient to cleave mutated substrates with preferential activity of Gly > Ala > Ser > Cys at the P1 position and Gly = Ala = Ser = Cys at the P2 position. These findings further supported the previous observation that a SUMO protein with similar residue substitutions at the Gly–Gly motif, such as directly-expressed human SUMO-1/3–AA, –GS and –GA mutants, could also undergo endogenous SUMOylation as effectively as the wild-type SUMO-1/3–GG protein [16]. These results together indicated that the Gly–Gly motif is not strictly required for SUMO precursor activation and SUMO protein modification.

### 2.5. Structural Modeling to Explore the Possible Mechanism of the Substrate Specificity of Ulp1

A clear correlation was observed between the residue size at the P1 and P2 positions and the cleavage efficiency. Therefore, structural modeling of the Ulp1–Smt3^P^ complex was performed to prompt the molecular understanding of the Ulp1-unique substrate specificity of Smt3 at the P1 and P2 positions (GG↓A motif). In the Ulp1–Smt3 complex, the Ulp1 catalytic triad C580–H514–D531 and residues W448, S513 and W515 jointly constructed a tight and shallow active tunnel, resembling a tapered structure wrapping around the Gly–Gly motif of Smt3 (Figure 4B–D). The side-chain-free diglycine motif retained extra free space, enabling the C-terminal tail to be able to flexibly pass through the active tunnel and, finally, place the P1–P1’ target bond above the active site without encountering any steric hindrance in the tunnel (Figure 4C,D). However, this changed when the Gly was replaced with residues possessing bulky side chains. When Gly at the P1 position was replaced with Ala, the side chain from Ala occupied more space in the Ulp1 substrate tunnel but still did not cause a significant blockage. Comparably, the -CH_2_OH side chain of Ser occupied the entire cavity and formed a serious conflict with the side chains of Ulp1–S513, –W448 and –C580, causing inefficient movement in the P1–P1’ peptide bond (GS↓A) in the active site for cleavage. Besides, the interaction also limited the orientation of S513 in the channel, thus greatly impairing the activity of Ulp1 against the Smt3^P^–GS↓A substrate (Figure 4C). For other residues with even bulkier side chains, such as Thr and Trp, the substrates could not even be docked in the substrate tunnel.

Different than at the P1 position, Ulp1 retained more free space at the P2 position to accommodate amino acids with certain side chains. Smt3^P2^ substrates with the substitution of Ala, Ser and Cys could all be easily introduced into the Ulp1 substrate pocket (Figure 4D). In the simulated structures, Gly and Ala could smoothly enter the substrate tunnel and retained some free space due to their small side chains. Ser and Cys have slightly bulkier side chains which almost occupied the cavity though this was without obvious conflict with the residues in the tunnel. Because of the ability of free orientation, these substrates could all be effectively cleaved by Ulp1. Comparably, when the P2 position was mutated to Thr, its large side chain group filled the substrate tunnel, making Thr difficult to move to the correct position for substrate cleavage. The bulky side chain of Thr may evoke obvious steric conflict to Ulp1 channel residues like S513 and W448 as well as to the catalytic residue H514 (Figure 4D).

Overall, the structural analysis indicated that it is the special taper-resembling active pocket of Ulp1 that conferred the distinguishing selectivity against residues of Smt3 at the P1 and P2 positions, in which only small amino acids without steric conflict with the Ulp1 substrate tunnel could flexibly adjust in the tunnel and approximate the active sites for proteolytic cleavage.

### 2.6. Gly in P1’ Position Expands Inclusivity of Ulp1 against Smt3

The structural analysis of the Ulp1–Smt3 complex indicated that extra moving space in the substrate tunnel was required to enable the P1–P1’ target peptide bond (GG↓A) to flexibly approach the nucleophilic residue C580 for cleavage. During this process, the side chain group of Ala at the P1’ position might form obvious steric hindrance between the Ulp1 catalytic residues of H514 and C580, thus affecting Ulp1 catalytic efficiency (Figure 5A). In the wild-type Smt3, the Gly–Gly motif in the substrate tunnel retained enough space to alleviate the obstruction through rotation or adjustment, so the C-terminus of Smt3 and the P1’ Ala were able to approach the catalytic pocket. Although, in the mutant Smt3^P^ substrates along with gradually-enlarged residues at the P1 and P2 position, such as Ala, Ser and Cys, the moving space in the substrate tunnel became limited, and the C-terminal tail was less flexible. Therefore, it was more difficult to decrease the steric hindrance of the P1’ position Ala from rotating. Based on this analysis, it can be speculated that the P1’ position Ala to Gly mutation may help expand Ulp1 inclusivity against the Smt3 residue at P1 and P2 positions because the side-chain-free Gly itself is highly-flexible, and P1 or P2 residues do not require more moving space to guarantee the high flexibility of the C-terminal tail.

To verify this speculation, Ulp1 specificity against the P1 and P2 sites of Smt3 when Smt3’s P1’ position was Ala or was mutated to Gly (Smt3^P^–XX↓A/G) was simultaneously characterized. When Ala was at the P1’ position, the cleavage efficiency of Ulp1 against the substrates Smt3^P^–GS↓A and Smt3^P^–GT↓A was 46.0% and 2.3%, respectively (Figure 5B). In comparison, when Gly was at the P1’ position, Ulp1 cleavage efficiency against the substrates Smt3^P^–GS↓G and Smt3^P^–GT↓G was increased to 99.1% and 94.88%, respectively. However, Ulp1 could not cleave the Smt3^P^–GI↓G substrate, indicating that this is a synergistic effect that the P1 site could only accommodate for amino acids with side chains that are less bulky than Ile, even when P1’ is Gly (Figure 5B). The specificity against the P2 position was also expanded. Compared to the fact that Ulp1 could only cleave substrates with a P2 residue smaller than Thr when Ala was in the P1’ position, the cleavage efficiency of Ulp1 against the Smt3^P^–TG↓G and Smt3^P^–WG↓G substrates was increased to 99.26% and 98.67%, respectively (Figure 5B). Not surprisingly, Ulp1 became able to accept more dual-mutations at P1–P2 positions when the P1’ position was Gly. The cleavage efficiency against the Smt3^P^–AA↓G substrate could be increased to 99.0% (Figure 5B), and Ulp1 also presented a good cleavage efficiency of 54.32% and 44.46% against the dual-mutation substrates Smt3^P^–AS↓G and Smt3^P^–SA↓G, respectively, which were not able to be cleaved when the P1’ position was Ala (Figure 5B). These results verified our speculation that the P1’ position Gly could expand Ulp1 inclusivity against the P1 and P2 positions of Smt3.

The effective cleavage of Ulp1 against bulky residues at the P1–P2 position, especially in the Smt3^P^–WG↓G substrates, was somewhat surprising, and the concern arose that the cleavage site might be shifted from Smt3^P^–WG↓GT to Smt3^P^–WGG↓T (the original P2’ residue of Smt3 is Thr). To validate whether the cleavage sites of Smt3^P^ substrates were shifted when P1’ was Gly, we mutated the original P2’ position Thr to Pro to generate the Smt3^P^–XX↓GP substrates. Pro was proven to be the blocking residue when it is at the P1’ position. If the cleavage site is not shifted, Ulp1 should show similar cleavage efficiency against both the Smt3^P^–XX↓GP and Smt3 ^P^–XX↓GT substrates. Otherwise, Smt3^P^–XXG↓P substrates would not be cleaved by Ulp1 at all. As shown in Figure 5B,C, Ulp1 showed a similar high-cleavage efficiency of approximately 99% against the Smt3^P^–GSGP, –GTGP and –AAGP substrates compared to that of the Smt3^P^–GSGT, –GTGT and –AAGT substrates, indicating that the cleavage site was still between P2P1↓GP/T in these substrates. The Smt3^P^–ASGT substrate showed a similar cleavage efficiency of 60% compared to that of the Smt3^P^–ASGP substrate, concluding that the cleavage site was also unshifted. In contrast, the Smt3^P^–TGGP and –WGGP substrates were almost unable to be cleaved while the Smt3^P^–TGGT and –WGGT substrates were fully cleaved, indicating that the cleavage site shifted from XX↓GP/T to XXG↓P/T in these substrates. One interesting observation was that the cleavage efficiency of the Smt3^P^–SAGP substrate had clearly decreased compared to that of the Smt3^P^–SAGT substrate, indicating a possible wobbling cleavage site in the Smt3^P^–SAGP substrate. Altogether, these results implied that a Gly at the P1’ position could expand the residue inclusivity of Ulp1 against the P1 position of Smt3^P^ from Gly > Ala > Ser > Cys to Gly = Ala = Ser = Cys = Thr. The expanded inclusivity of Ulp1 against huge residues in the P2 position of Smt3^P^ might be attributed to the shifted cleavage site from the original P1–P1’ to the P1’–P2’ position.

A previous work demonstrated that the SUMO P1’ position residue is involved in discriminating the activity of the human SUMO protease SENP1 on SUMO 1, SUMO 2 and SUMO 3 isoforms [17]. Despite the fact that a detailed mechanism is unclear, the special significance of the P1’ position residue in the SUMO protein is worth being further explored. Here, our results revealed the essential role of the residue Ala at the P1’ position of Smt3 in maintaining Ulp1 specificity in the P1 and P2 sites of Smt3, that is, restricting only small amino acids with free space in the Ulp1 substrate tunnel can be flexibly rotated to decrease the steric hindrance of Ala in the active residues H514 and C580 and helping the P1–P1’ target peptide bond approach the nucleophilic residue C580 for cleavage. The P1, P2 and P1’ residues of Smt3 collectively conferred Ulp1’s unique specificity for the Smt3 precursor process. This unique mechanism, applied by the Ulp1–Smt3 complex, may help guarantee that all SUMO proteins are accurately cleaved behind the GG↓ motif because the C-terminal fragment of the wild-type SUMO protein rarely contains the other small amino acids that we characterized here, such as Ala, Ser and Cys, which reflects the wisdom of the eukaryotes in evolving the SUMO system.

## 3. Materials and Methods

### 3.1. Vector Construction

Genes encoding *Saccharomyces cerevisiae* SUMO protease Ulp1 (403–621) and SUMO protein Smt3 were amplified from EBY100 *(Ura*^+^, *leu*^−^*, trp*^−^*)* genome. Smt3^I^ and Smt3^P^ mutants (including Smt3–XG↓A, Smt3–GX↓A, Smt3–XX↓A and Smt3–GX↓G, Smt3–XG↓G, Smt3–XX↓G) were obtained using Smt3 as a template. Then, Ulp1 protease and the divergent substrates were cloned into previously published centromeric vector pESD to generate Ulp1-GAL1-GAL10-Aga2-FLAG-Smt3^X^-HA expression cassettes (Figure 1). The detailed construction principle can be referred to in a previous publication [18].

### 3.2. YESS–PSSC Analysis

Recombinant plasmids containing Ulp1 and its divergent substrates were transformed into *S. cerevisiae* strain EBY100 competent cells and then plated on YNB-Galactose-CAA selective plates (6.7 g/L YNB, 5 g/L casamino acids, 20 g/L glucose and 15 g/L agar). Transformed cells were cultivated in YNB-Glucose-CAA medium to an OD600 of 4.0–5.0, followed by induction with an initial OD600 of 0.5 in YNB-Galactose-CAA medium. After induction for 8 h at 30 ℃, approximately 10^6^ cells were harvested by centrifugation and washed once with 200 μL cell washing Buffer A (1 × PBS buffer, 0.5% bovine serum albumin (BSA) and 1mM ethylenediaminetetraacetate (EDTA), pH 7.4) and twice with 200 μL cell washing Buffer B (1X PBS buffer, 0.5% BSA, pH7.4). The washed cells were then incubated with 0.1 μM anti-HA-FITC antibody and 0.1 μM anti-FLAG-iFlor647 antibody (GenScript, Nanjing, China) for 15 min in dark, followed by being washed twice with 1× PBS buffer to completely remove the unbound antibodies. Subsequently, the antibody-labeled cells were resuspended in 400 µL 1× PBS buffer for the fluorescence analysis using Beckman Coulter CytoFLEX Flow Cytometer (Beckman Coulter, USA). The anti-FLAG-iFluor 647 and anti-HA-FITC fluorescent intensity were, respectively, detected with the APC channel (660/20 nm BP) and FITC channel (525/40 nm BP). The cleavage efficiency was calculated as: [Normalized cleavage efficiency] = ([Cells exhibiting only iFluor 647 fluorescence signals]/([Cells exhibiting only iFluor 647 fluorescence signals] + [Cells exhibiting both iFluor 647 and FITC fluorescence signals]) × 100%.

### 3.3. Structure Modeling

The X-ray crystal structure of Ulp1 protease domain in complex with Sumo protein (Smt3) was downloaded from protein data bank (PDB: 1euv). Structural modeling of Smt3 mutants was performed using the ZDOCK program in Discovery Studio (DS) software [28], setting the angular step size parameter (6°) and RMSD Cutoff (6.0 Å) to get ZDOCK scores and clustering results for these conformations. The RDOCK protocol was then used to refine and reorder the poses of docking structure.

## 4. Conclusions

Here, we utilized the YESS–PSSC strategy to profile the substrate specificity of the SUMO protease Ulp1. Through combining structural modeling analyses, we systematically deciphered the residue specificity of the *S. cerevisiae* SUMO protease Ulp1 against the P2, P1 and P1’ positions of Smt3 and demonstrated that the Gly–Gly motif at the P2–P1 positions is not strictly required for Smt3 precursor activation. Furthermore, based on further structural analysis and experimental verification, we preliminarily revealed that the P1, P2 and P1’ residues of Smt3 collectively maintained Ulp1’s accurate cleavage behind the Gly–Gly motif of Smt3. These results might promote the further understanding of the substrate recognition of other SUMO proteases and facilitate the design and discovery of the inhibitor for the treatment of abnormal SUMOylation-associated diseases.

## Figures and Tables

**Figure 1 ijms-23-12188-f001:**
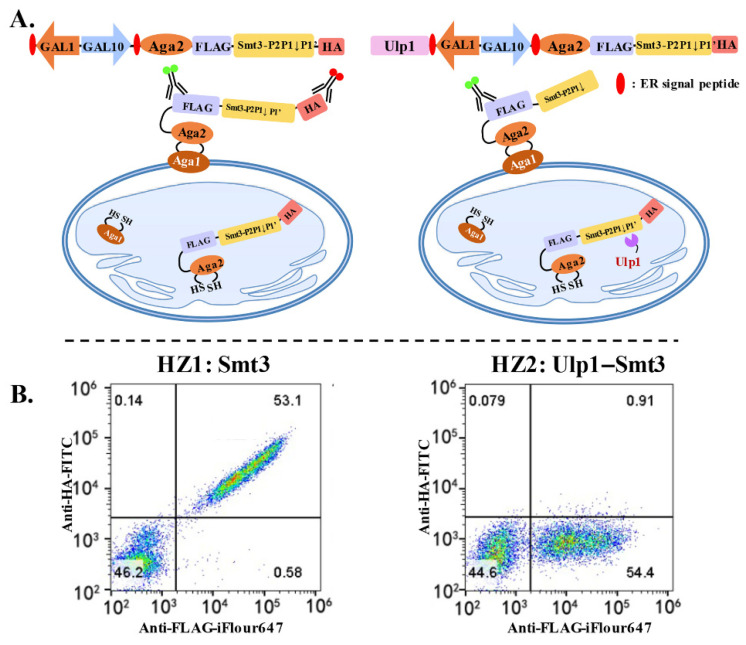
YESS–PSSC system for quantitatively characterizing Ulp1–Smt3 specificity. (**A**) Scheme of YESS–PSSC system for substrate specificity analysis of Ulp1 against Smt3. Left: represents cells merely expressing substrate part (Aga2-FLAG-Smt3-HA). Right: represents cells simultaneously expressing Ulp1 and substrate part (Aga2-FLAG-Smt3-HA). (**B**) Flow cytometry analysis that characterizes Ulp1–Smt3 specificity in YESS–PSSC system. Left: HZ1 corresponded to the result of (**A**) left. Right: HZ2 corresponded to the result of (**A**) right.

**Figure 2 ijms-23-12188-f002:**
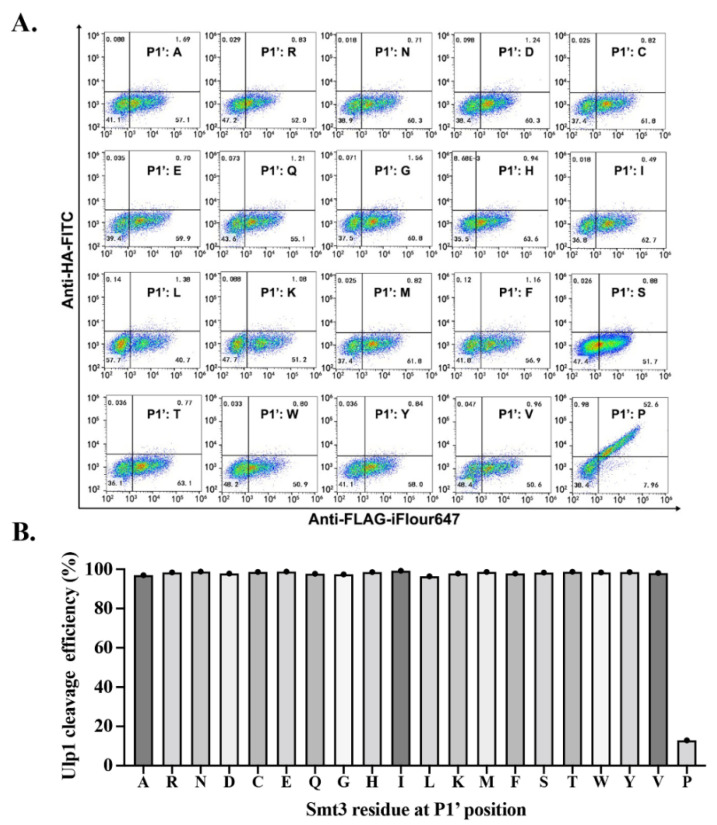
Mapping residue specificity of Ulp1 at Smt3 P1’ position. Ulp1-GAL1-GAL10-Aga2-FLAG-Smt3GG↓X-HA (X is an arbitrary amino) cassettes are, respectively, expressed in YESS–PSSC system, followed by direct flow cytometry analysis. (**A**) Representative flow cytometry results. (**B**) Data of A are presented as average cleavage efficiency (*n* = 3 independent experiments).

**Figure 3 ijms-23-12188-f003:**
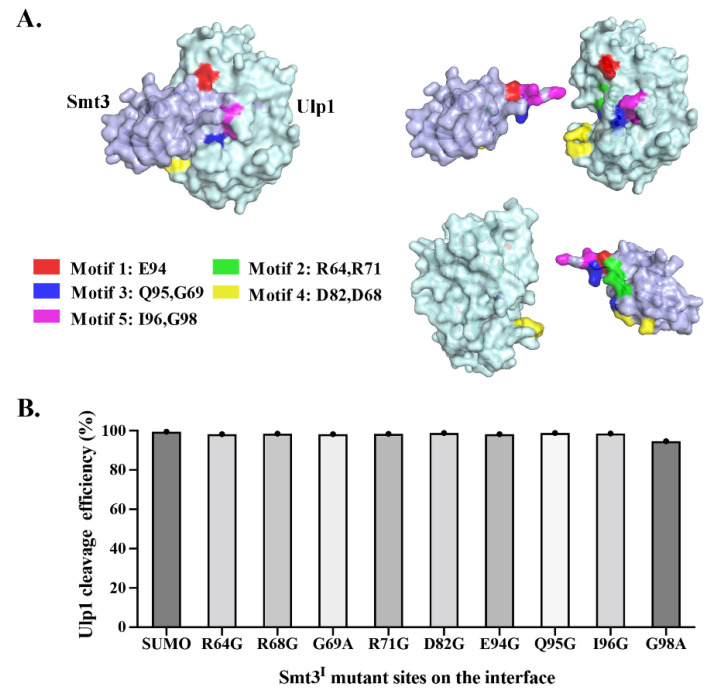
Analysis of Ulp1 against Smt3^I^ mutants. (**A**) Interface of Ulp1–Smt3 in the complex structure (PDB,1euv). The motif and corresponding residues in Smt3 have been labeled with different colors. (**B**) Evaluation of Ulp1 cleavage against different Smt3 variants in YESS–PSSC system. Cleavage activity of Ulp1 against Smt3^I^ substrates are presented as average cleavage efficiency (*n* = 3 independent experiments).

**Figure 4 ijms-23-12188-f004:**
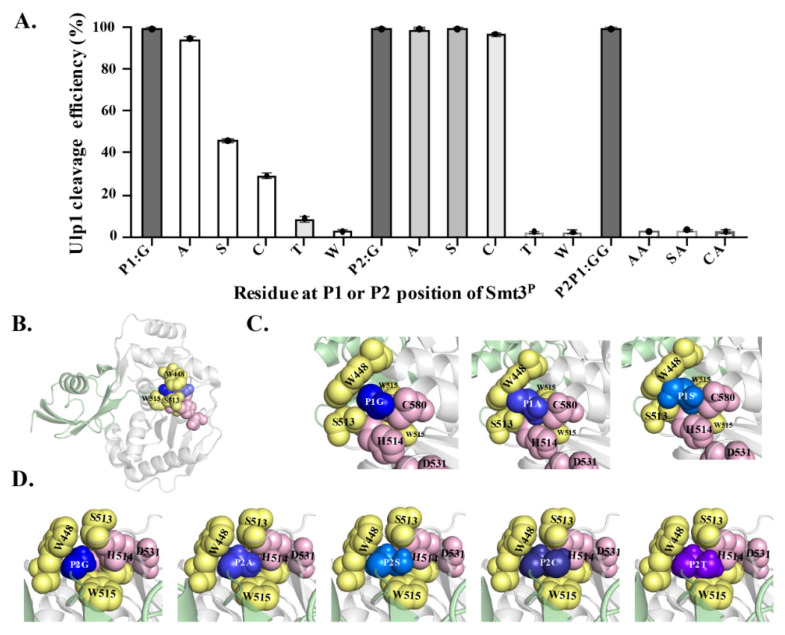
Analyzing the residue specificity of Ulp1 against Smt3 at P1 and P2 positions. (**A**) Gly–Gly at P1 and P2 position of Smt3 were mutant to arbitrary amino acids (GX_1_↓, X_2_G↓ and X_2 × 1_↓), with the mutant alleles named Smt3^P^. Cleavage efficiency of Ulp1 against Smt3^P^ alleles are presented as average cleavage efficiency (*n* = 3 independent experiments). (**B**) Structure of Ulp1–Smt3 complex (PDB, 1euv). Grey represents protease Ulp1 whose substrate pocket is jointly-constructed by catalytic triad C580–H514–D531(pink) and residues of W448, S513 and W515 (yellow). Light green represents Smt3, and purple and blue represent Gly at P1 and P2 position, respectively. (**C**) Structural modeling of Ulp1–Smt3 P1 position mutants, including Smt3–GG↓, –GA↓ and –GS↓. (**D**) Structural modeling of Ulp1–Smt3 P2 position mutants, including Smt3–GG↓, –AG↓, –SG↓, –CG↓ and –TG↓.

**Figure 5 ijms-23-12188-f005:**
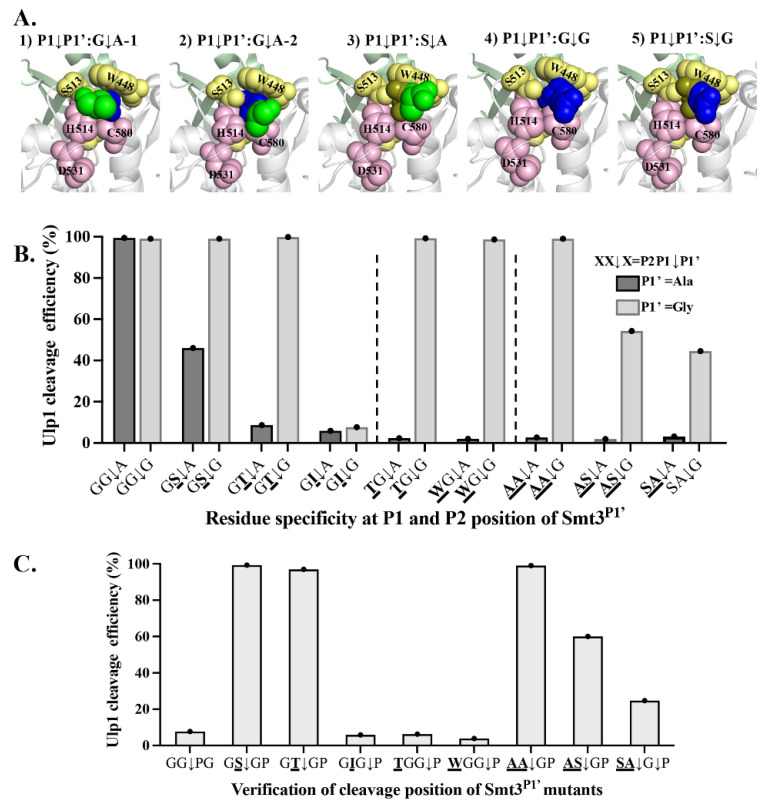
P1’ position Gly expands residue inclusivity of Ulp1 against P1 and P2 position of Smt3. (**A**) The substrate pocket structure of Ulp1–Smt3 and Ulp1–Smt3^P1P1^^’^ complex at P1 and P1’ position. Blue is P1 position Gly; green is P1’ position Ala; and brown is P1 position Ser. (1) The first conformation of Smt3–GG↓A substrate’s target peptide bond between P1–P1’ position (GG↓A) requires enough flexibility to approach nucleophilic residue C580 for cleavage. (2) The second conformation of Smt3–GG↓A substrate’s P1’ position Ala side chain extension may cause obstruction in Ulp1 catalytic residues H514 and C580 ability to affect cleavage efficiency. (3) P1 position amino acids with longer side chains, like Ser, would limit flexibility of C-terminal tail in the tunnel, which may prevent P1’ position Ala from rotating to remove the steric hindrance in Ulp1 catalytic residues, and the P1–P1’ peptide bond was not close to catalytic residue C580. Therefore, Ulp1 cleaving efficiency was affected. (4) P1’ position Gly has high flexibility and does not form hindrance to Ulp1 catalytic residues H514, C580 and D531. (5) P1 or P2 position residues, like Ser, do not require a huge free space to guarantee high flexibility of P1’ position Gly. (**B**) The residue specificity of Ulp1 on Smt3^P^ when P1’ position was Ala or Gly. P2P1↓P1’ position residue in wild-type Smt3 is GG↓A. (**C**) Verification of cleavage position in Smt3^P^ substrates when P1’ was Gly. P2P1∣P1’P2’ position residue in wild-type Smt3 is GG↓AT. Cleavage efficiency of Ulp1 against corresponding substrates is presented as average cleavage efficiency of three independent experiments.

## Data Availability

Not applicable.

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
