# Peer review of "Profiling Substrate Specificity of the SUMO Protease Ulp1 by the YESS–PSSC System to Advance the Conserved Mechanism for Substrate Cleavage"

_ijms, 2022, doi:10.3390/ijms232012188_

Round 1
Reviewer 1 Report
The authors explore the requirements of the SMT3 C-terminus for cleavage by the ULP1 enzyme. They find that the C-terminal GG motif conserved in the SUMO family is not strictly required for cleavage. They also make the surprising observation that a very small residue (Gly) following the GG-motif makes the the positions preceding the cleavage site much more permissive for larger residues.
First, the language has to be improved substantially. In the current version, many sentences just don't make sense, to a degree that impedes the entire reviewing process.
As for the content, I strongly suggest that the authors validate the more suprising results of their analysis by a conventional cleavage assay. While the YESS-PSSC method has its advantages, it works at rather unphysiological conditions (in the ER) and cannot detect the position of the cleavage (see below for specifics)
Some of the results are noteworthy, but not totally unexpected. The fact that the S1 and S2 positions can tolerate non-glycine residues is clearly of interest, but does not differ fundamentally from the situation in ubiquitin. I find it particularly interesting that the P2 position appears to be more stringent than the P1 position (figure 4a). Recently, a 'gatekeeper motif' was identified in all major DUB classes, which prevents large residues a the P2 position (https://pubmed.ncbi.nlm.nih.gov/32719160). This motif is also present in the CE-clan, of which ULP1 is a member. Specifically, W515 of ULP1 is the crucial aromatic residue for the gatekeeper function.
A really surprising finding is the permissiveness of the P2 position under conditions where P1' is occupied by a glycine (Fig 5b). Even a big residue like Trp is reported to allow cleavage after WG|G. I find this result so unexpected that I would really like to see it reproduced by an idependent method. Can we really be sure that this substrate is not cleaved after WGG| ? In the YESS-PSSC assay, such a shift would go unnoticed. One possibility would be to detect the C-terminal residue of the cleavage product on the yeast cell. Another possibility would be an in vitro cleavage reaction with MS-analysis of the cleavage product.
As a minor point, Figure 2B could use some axis labels.
Author Response
To Review 1:
First of all, we are very grateful to you for carefully reviewing our manuscript and providing us these professional comments, which are very helpful to the improvement of our manuscript.
The detailed point-to-point response to all comments are as follows.
- First, the language has to be improved substantially. In the current version, many sentences just don't make sense, to a degree that impedes the entire reviewing process.
Reply 1: We have sought help from English native speaker to improve the writing of this manuscript. As a result, most of the sentences are revised and highlighted in the re-submitted version.
- As for the content, I strongly suggest that the authors validate the more surprising results of their analysis by a conventional cleavage assay. While the YESS-PSSC method has its advantages, it works at rather unphysiological conditions (in the ER) and cannot detect the position of the cleavage (see below for specifics)
Reply 2: Thanks for your suggestion. Although YESS-PSSC method cannot directly detect the cleavage position, additional experiments based on this method can help us deduce the cleavage site of Ulp1 on mutant Smt3P1’ substrates. Specific results are shown at Reply 4.
- Some of the results are noteworthy, but not totally unexpected. The fact that the S1 and S2 positions can tolerate non-glycine residues is clearly of interest, but does not differ fundamentally from the situation in ubiquitin. I find it particularly interesting that the P2 position appears to be more stringent than the P1 position (figure 4a). Recently, a 'gatekeeper motif' was identified in all major DUB classes, which prevents large residues a the P2 position (https://pubmed.ncbi.nlm.nih.gov/32719160). This motif is also present in the CE-clan, of which ULP1 is a member. Specifically, W515 of ULP1 is the crucial aromatic residue for the gatekeeper function.
Reply 3: Recognition of C-terminal Gly-Gly motif is a typical feature of SUMO proteases and DUBs proteases. Despite no sequence or structural similarity, both of them seem utilize the similar ‘aromatic gate’ motif to guarantee their high specificity towards Gly-Gly motif. In Ulp1, W448 and W515 are mainly responsible for clamping down P1-Gly and P2-Gly in the active tunnel (Figure 4B, C, D). Although this exquisite structure looks like can prevent larger residues insertion in the tunnel, our experiment verified that the mutated Smt3 substrates can be effectively cleaved by Ulp1, with preferential activity of Gly>Ala>Ser>Cys at P1 position and Gly=Ala=Ser=Cys at P2 position. In our view, experimental results are more reliable than the theoretical predication based on structural analysis. We hope our initial exploration about substrate specificity of Ulp1 can attract more researchers to clearly study the detailed specificality of other SUMO proteases and DUBs proteases in the future.
- A really surprising finding is the permissiveness of the P2 position under conditions where P1' is occupied by a glycine (Fig 5b). Even a big residue like Trp is reported to allow cleavage after WG|G. I find this result so unexpected that I would really like to see it reproduced by an independent method. Can we really be sure that this substrate is not cleaved after WGG|? In the YESS-PSSC assay, such a shift would go unnoticed. One possibility would be to detect the C-terminal residue of the cleavage product on the yeast cell. Another possibility would be an in vitro cleavage reaction with MS-analysis of the cleavage product.
Reply 4: Thanks for your professional and constructive comments. This suggestion is extremely critical for our article. In fact, the surprising capacity of Ulp1 on Smt3P1’ P2 position residues also arose our concern that cleavage site might be shifted from P2P1↓GT to P2P1G↓TY (T are the natural residues after P1’ position in Smt3). Theoretically, Ulp1 should maintain high specificity and accurate cleavage at P2P1↓AT site towards Smt3 substrates. However, when exploring the effect of Smt3 P1’ position A-G mutation to Ulp1, Smt3P1’ C-terminal flexibility was artificially increased, which may affect the high specificity of Ulp1. Therefore, during submitting our manuscript, we have performed another experiment, which was sufficient to validate whether the cleaving position was shifted from P2P1↓GT to P2P1G↓T site. “Verification of cleaving position of Smt3P1’ mutants” was added as Figure 5C and corresponding context has been described in Line 315-359 in the revised manuscript. The specific principle and results are as follows:
“The effective cleavage against bulky residues at P1-P2 position, especially the Smt3P1’-WG↓G substrates, was a little surprising, which arose the concern that cleavage site might be shifted. Therefore, we performed further characterization on substrate with either Pro or Thr at P2’ position. Pro was proven to be the blocking residue when it is at the P1’ position and Thr is the original residue at the P2’ position in Smt3. To validate whether the cleaving site of Smt3P1’ substrates were shifted, we mutated Thr at P2’ position of the Smt3P1’C-terminal into Pro to generate the Smt3P1’-P2P1↓GP substrates. If the cleaving site is not shifted, Ulp1 should show similar cleavage efficiency against the Smt3P1’-P2P1↓GP and Smt3P1’-P2P1↓GT substrates. Otherwise, Smt3P1’-P2P1G↓P substrates will not be cleaved by Ulp1. As shown in Figure 5B, C, Ulp1 showed similar high cleavage efficiency against Smt3P1’-GSGP, GTGP and AAGP substrates as well as Smt3P1’-GSGT, GTGT and AAGT substrates of approximately 99%, indicating the cleaving site was between P2P1↓GP/T in these substrates. Unshifted cleavage site could be concluded in the Smt3P1’-ASGP and Smt3P1’-ASGT substrates, as similar cleavage efficiency of 60% was obtained. In contrast, Smt3P1’-TGGP and WGGP substrates were almost unable to be cleaved while almost full cleavage against Smt3P1’-TGGT and WGGT substrates was obtained, indicating the cleaving site shifted from P2P1↓GT to P2P1G↓P/T in these substrates. One interesting observation was that cleavage efficiency against Smt3P1’-SAGP substrate was clearly decreased comparing to that of Smt3P1’-SAGT substrate, indicating a possible wobbling cleavage site in Smt3P1’-SAGP substrate. Taken together, these results implied that Gly at P1’ position could expand residue inclusivity of Ulp1 against Smt3P1’ P1 position from Gly>Ala>Ser>Cys to Gly=Ala=Ser=Cys=Thr. The expanded inclusivity of Ulp1 against Smt3P1’ P2 position to huge residues might more because the shifted cleaving site from original P1-P1’ to P1’-P2’ position. It could be concluded that only small residues could be accommodated in the P1 and P2 positions to render an effective cleavage, but the overall cleavage efficiency is synergistically decided by residues in P2-P1-P1’ positions.”
As a minor point, Figure 1B could use some axis labels.
Reply 5: Thanks for your comments. We have used axis labels in Figure 1B.

Reviewer 2 Report
"Profiling substrate specificity of SUMO protease Ulp1 by YESS-PSSC system to advance the conserved mechanism for substrate cleavage" by Zhang et al has described an in vivo approach based on yeast endoplasmic sequestration screening to assay SUMO cleavage activity of the SUMO protease Ulp1 in Saccharomyces cerevisiae. This work provides an easy and quick methodology to test Smt3-maturating activity of the enzyme. The authors also used fluorescence-based assay to distinguish an intact and processed forms of Smt3 to monitor the enzyme activity. Using the assay, they also analyzed mutant variants of Smt3 aiming to understand the importance of amino acid residues of the substrate. I found that the assay is probably useful for Sumo field of study and the authors have written in an easy way to follow. I, however, found certain experiments may be controversial or need controls and/or discussion. Here are major and minor points required to be addressed.
Major points:
- Does the EBY100 strain have wild-type ULP1 gene? How the endogenously expressed Ulp1 might affect to the YESS-PSSC assay used in this study? Is the use of delta-ulp1 strain more appropriate? Please explain and discuss.
- Results and discussion (2.3): are there any scientific reasons why some variants are mutated to Glycine, while some to Alanine? Do the authors think the discrepancy between these two amino acids could be omitted in such a case? Please explain and discuss.
- Results and discussion (2.3): Line 204, could the authors clarify how ‘slightly lower’ mean in terms of number (%)? Are the result of G98A in Fig. 3B and P1:A in Fig. 4A theoretically the same? In my copy, they looked different; the latter was lower than the former.
- Results and discussion (2.4): the authors cited Yamada et al (2012) [Ref. 16], which shows a differential conjugation efficiency of Sumo variants in human Hela cells. I found the results in their paper and the current manuscript are significantly contradicting. While (P2P1:AA) variant in Fig 4A of the current manuscript showed no activity at all, the AA of both Sumo-1 and -2 can still form conjugates, suggesting that maturation of both occurs. Moreover, in the current manuscript, P2P1:GA and P2P1:AG variants can be cleaved as wild-type (Fig 4A), these variants are non-functional in case of Sumo1/3. The current version of this manuscript failed to discuss the discrepancy; thereby the conclusion that ‘the diglycine motif may be not an essential feature of Sumo proteins’ is premature.
- The authors showed an interesting observation in Figure 5B that mutating the P1’ position to glycine significantly increased the cleavage activity. Given that the tail of Smt3 is flexible and long enough to slightly move around, theoretically when the P1’ is mutated to G and the amino acid upstream of it (P1) is small, it is possible that the cleavage site could be shifted by one amino acid to the C-terminus, i.e. to P2P1P1’↓P2’ instead of P2P1↓P1’P2’. This is an important point, especially for the current manuscript, because it may totally change the conclusion of the paper that the amino acid downstream of cleavage site contributes to the cleavage efficiency. Additional experiment is required in order to be conclusive.
Minor points:
- Full form of YESS and PSSC must be clarified in the text. The principle of the approach must be briefly explained, too.
- Figure 1: a schematic representation of Smt3’s [P2, P1↓P1’] would be helpful for an easier understanding.
- Figure 1A: it might be easier to follow if the left and right panels of the figure 1A are described separately in the legend.
- Figure 1B: some part of the legend of figure 1B should be moved to or explained in the Results and discussion part. Though it is well explained in the methodology, it is slightly hard to follow from the current version. For example, what do the FLAG/HA tags and the fluorescence signals of FITC/iFluor647 imply?
- Figure 1B: please label the X-axis and Y-axis. (Anti-FLAG, Anti-HA etc.)
- Figure 3A: Labels should be added. Which one is the Smt3, which Ulp1?
- Materials and methods: is the pESD vector centromeric or integrative? If integrative, please add the information of what restriction enzyme was used to cleave before integration.
- Materials and methods: Smt3x must be described. Is it the same as Smt3P?
- Materials and methods: the genotype of EBY100 described in 3.2 should be described in 3.1 instead.
- Results and discussion: Line 177, should ‘Our similar results with …’ be ‘Our results similar to …’?
- Results and discussion (2.4): it is a bit strange in my opinion to mention Figure 4B before Figure 4A.
- Results and discussion (2.4): line 247, should ‘Regrettably’ be ‘As expected’ since the di-glycine motif is highly conserved among eukaryotes and mutation of it should affect its maturation and cleavage?
- Figures 2B, 3B, 4A, 5B: some showed error bars, while some did not. What do the bars represent (SD, SE, or else)?
- Little to moderate change of English grammar and spelling check are required. For example, Line 38: their maturation require à their maturation requires; Line 46: is benefit à is beneficial; Line 397, was download à was downloaded, etc.
Author Response
To Review 2:
First of all, we are very grateful for your careful review of our manuscript. We think your opinions are very helpful for the improvement of our manuscript.
The detailed point-by-point response to all comments are as follows.
Major points:
- Does the EBY100 strain have wild-type ULP1 gene? How the endogenously expressed Ulp1 might affect to the YESS-PSSC assay used in this study? Is the use of delta-ulp1 strain more appropriate? Please explain and discuss.
Reply 1: Thanks for your professional comments. S. cerevisiae EBY100 strain used in this study has wild-type ULP1 gene and could expressed during the assay. But endogenous wild-type Ulp1 would not interfere and cleave our testing substrate (Aga2-FLAG-Smt3GG↓-HA), because the substrate part was escorted into yeast endoplasmic reticulum by ER signal peptide while endogenous wild-type Ulp1 is usually localized at the nucleus guided by its N-terminal nuclear localization signal (NLS) (Line 118-123). Corresponding result has been shown in the left part in Figure 1B.
In addition, knocking out of ULP1 gene has been demonstrated to result in a lethal growth phenotype to yeast, which cannot be complemented by Ulp1 C-terminal protease domain (403-621aa) (Ref 15 in manuscript). Since endogenous wild-type Ulp1 would not interfere our results, so the EBY100 strain is proper for our testing process.
- Results and discussion (2.3): are there any scientific reasons why some variants are mutated to Glycine, while some to Alanine? Do the authors think the discrepancy between these two amino acids could be omitted in such a case? Please explain and discuss.
Reply 2: Thanks for your professional comments. We don’t think the discrepancy between these two amino acids could be omitted. Here, seven residues of the 9 sites are originally not glycine while two are glycine. Thus, seven residues were mutated into glycine, with simpleststructure and property. Two sites including G69 and G98 were mutated to alanine, the second simplest amino acid.
- Results and discussion (2.3): Line 204, could the authors clarify how ‘slightly lower’ mean in terms of number (%)? Are the result of G98A in Fig. 3B and P1: A in Fig. 4A theoretically the same? In my copy, they looked different; the latter was lower than the former.
Reply 3: Thanks for your professional comments. According to your suggestion, we have described this result with specific number in the revised manuscript (Line 180-182, 191-194). Specifically, the cleavage efficiency of eight Smt3 interface mutants was all about 98%, excepting Smt3I-G98A mutant whose cleavage efficiency was 94.65%. Therefore, SUMO proteins (GA98th↓A) exhibited cleavage efficiency about 5% lower than wildtype Smt3 and other mutations. Besides, the result of G98A in Fig. 3B and P1: A in Fig. 4A theoretically are the same. The slight difference between these two figures is caused by the inevitable error in two batches of independent experiments, which doesn’t affect our conclusion.
- Results and discussion (2.4): the authors cited Yamada et al (2012) [Ref. 16], which shows a differential conjugation efficiency of Sumo variants in human Hela cells. I found the results in their paper and the current manuscript are significantly contradicting. While (P2P1: AA) variant in Fig 4A of the current manuscript showed no activity at all, the AA of both Sumo-1 and -2 can still form conjugates, suggesting that maturation of both occurs. Moreover, in the current manuscript, P2P1:GA and P2P1:AG variants can be cleaved as wild-type (Fig 4A), these variants are non-functional in case of Sumo1/3. The current version of this manuscript failed to discuss the discrepancy; thereby the conclusion that ‘the diglycine motif may be not an essential feature of Sumo proteins’ is premature.
Reply 4: In the cited reference published by Yamada et al (2012), we guess mature SUMO-1/3-AA/GS/GN/GA↓ mutants are directly generated and expressed in Hela cells, which means they don’t require to be activated by SUMO protease before SUMOylating intracellular proteins. Therefore, the result of Smt3-AA mutants cannot be cleaved by Ulp1 in our manuscript doesn’t contradict with the results of Sumo-1/3 AA can still form conjugates in [Ref. 16]. The specific description is ‘To determine the importance of the GG motif in the SUMO modification pathway, first we performed PCR to generate a series of GG motif mutants of SUMO-1 and SUMO-3 using appropriate DNA primers (primer sequences available upon request). The mutated constructs were then cloned into pEGFP vector (Clontech, Takara Bio, Kusatsu, Japan), resulting in pEGFP- SUMO-1/3-GG, -AA, -GA, -GS, -GN, -GF, -GL, -GP, -GT, -GY, -GV, -GK, -GD, -GE, -GC, -GW, -GR, -GQ, -GH, -GI, -GM, and -GΔG. In the case of SUMO-1, the pEGFP-SUMO-1-AG mutant was also generated. These constructs were transfected into HeLa cells, which were then cultured for 24h’.
In addition, processing of SUMO precursor by SUMO protease and conjugation of mature SUMO to target proteins by SUMO activating enzyme-conjugating enzyme-ligase are two independent processes. Our work was designed to dissect the substrate specificity of Ulp1 as well as reveal the possible mechanism for its specificity. Whether these mutants can SUMOylate intracellular proteins need be validated by additional experiments. So, as you suggested, we have deleted our premature conclusion of ‘the diglycine motif may be not an essential feature of Sumo proteins’ in the revised manuscript.
- The authors showed an interesting observation in Figure 5B that mutating the P1’ position to glycine significantly increased the cleavage activity. Given that the tail of Smt3 is flexible and long enough to slightly move around, theoretically when the P1’ is mutated to G and the amino acid upstream of it (P1) is small, it is possible that the cleavage site could be shifted by one amino acid to the C-terminus, i.e. to P2P1P1’↓P2’ instead of P2P1↓P1’P2’. This is an important point, especially for the current manuscript, because it may totally change the conclusion of the paper that the amino acid downstream of cleavage site contributes to the cleavage efficiency. Additional experiment is required in order to be conclusive
Reply 5: Thanks for your professional and constructive comments. This suggestion is extremely critical for our article. In fact, we also suspected that the surprising capacity of Ulp1 on Smt3P1’ P2 position residues was caused by the shift of cleaving position from P2P1↓GT to P2P1G↓TY (T are the natural residues after P1’ position in Smt3). Theoretically, Ulp1 should maintain high specificity and accurate cleavage at P2P1↓AT site towards Smt3 substrates. However, when exploring the effect of Smt3 P1’ position A-G mutation to Ulp1, Smt3P1’ C-terminal flexibility was artificially increased, which may affect the high specificity of Ulp1. Therefore, during submitting our manuscript, we have performed another experiment, which was sufficient to validate whether the cleaving position was shifted from P2P1↓GT to P2P1G↓T site. “Verification of cleaving position of Smt3P1’ substrates when P1’ was Gly” was added as Figure 5C and corresponding context has been described in Line 315-359 in the revised manuscript. The specific principle and results are as follows:
“The effective cleavage against bulky residues at P1-P2 position, especially the Smt3P1’-WG↓G substrates, was a little surprising, which arose the concern that cleavage site might be shifted. Therefore, we performed further characterization on substrate with either Pro or Thr at P2’ position. Pro was proven to be the blocking residue when it is at the P1’ position and Thr is the original residue at the P2’ position in Smt3. To validate whether the cleaving site of Smt3P1’ substrates were shifted, we mutated Thr at P2’ position of the Smt3P1’C-terminal into Pro to generate the Smt3P1’-P2P1↓GP substrates. If the cleaving site is not shifted, Ulp1 should show similar cleavage efficiency against the Smt3P1’-P2P1↓GP and Smt3P1’-P2P1↓GT substrates. Otherwise, Smt3P1’-P2P1G↓P substrates will not be cleaved by Ulp1. As shown in Figure 5B, C, Ulp1 showed similar high cleavage efficiency against Smt3P1’-GSGP, GTGP and AAGP substrates as well as Smt3P1’-GSGT, GTGT and AAGT substrates of approximately 99%, indicating the cleaving site was between P2P1↓GP/T in these substrates. Unshifted cleavage site could be concluded in the Smt3P1’-ASGP and Smt3P1’-ASGT substrates, as similar cleavage efficiency of 60% was obtained. In contrast, Smt3P1’-TGGP and WGGP substrates were almost unable to be cleaved while almost full cleavage against Smt3P1’-TGGT and WGGT substrates was obtained, indicating the cleaving site shifted from P2P1↓GT to P2P1G↓P/T in these substrates. One interesting observation was that cleavage efficiency against Smt3P1’-SAGP substrate was clearly decreased comparing to that of Smt3P1’-SAGT substrate, indicating a possible wobbling cleavage site in Smt3P1’-SAGP substrate. Taken together, these results implied that Gly at P1’ position could expand residue inclusivity of Ulp1 against Smt3P1’ P1 position from Gly>Ala>Ser>Cys to Gly=Ala=Ser=Cys=Thr. The expanded inclusivity of Ulp1 against Smt3P1’ P2 position to huge residues might more because the shifted cleaving site from original P1-P1’ to P1’-P2’ position. It could be concluded that only small residues could be accommodated in the P1 and P2 positions to render an effective cleavage, but the overall cleavage efficiency is synergistically decided by residues in P2-P1-P1’ positions.”
Minor points:
- Full form of YESS and PSSC must be clarified in the text. The principle of the approach must be briefly explained, too.
Reply 6: Thanks for your suggestion. We have added the full form of YESS-PSSC system in the revised manuscript, that is Yeast ER sequestration screening (YESS) based protease−substrate specificity characterization (PSSC). Besides, brief principle about YESS-PSSC system has been explained (Line 129-135, 400-403).
- Figure 1: a schematic representation of Smt3’s [P2, P1↓P1’] would be helpful for an easier understanding.
Reply 7: Thank you very much. We have corrected the schematic diagram according to your suggestion.
- Figure 1A: it might be easier to follow if the left and right panels of the figure 1A are described separately in the legend.
Reply 8: Thank you very much. We have revised the legend of Figure 1A according to your suggestion.
- Figure 1B: some part of the legend of figure 1B should be moved to or explained in the Results and discussion part. Though it is well explained in the methodology, it is slightly hard to follow from the current version. For example, what do the FLAG/HA tags and the fluorescence signals of FITC/iFluor647 imply?
Reply 9: Thanks for your suggestion. We have moved the explanation about Figure 1B in the section of Results and discussion. FITC and iFluor647 fluorescence signals have been corrected as anti-HA-FITC and anti-FLAG-iFluor647 fluorescence signals. In addition, we have further refined the explanation about methodology for readers’ better understanding (Line 129-135).
- Figure 1B: please label the X-axis and Y-axis. (Anti-FLAG, Anti-HA etc.)
Reply 10: Thanks for your suggestion. We have added the label in Figure 1B.
- Figure 3A: Labels should be added. Which one is the Smt3, which Ulp1?
Reply 11: We really appreciate your suggestion. Labels of Smt3 and Ulp1 have been added in the legend of Figure 3A.
- Materials and methods: is the pESD vector centromeric or integrative? If integrative, please add the information of what restriction enzyme was used to cleave before integration.
Reply 12: Thanks for your suggestion. pESD is centromeric plasmid and can keep stable without integration. We supplemented this information in the revised manuscript (Line 381).
- Materials and methods: Smt3x must be described. Is it the same as Smt3P?
Reply 13: Thanks for your suggestion. We have refined the description as ‘Smt3P alleles, including Smt3-NG, Smt3-GN and Smt3P1’G-GN, Smt3P1’G-NG mutants’ for better understanding.
- Materials and methods: the genotype of EBY100 described in 3.2 should be described in 3.1 instead.
Reply 14: Thanks for your suggestion. We have moved the description about genotype of EBY100 to 3.1 (Line 378).
- Results and discussion: Line 177, should ‘Our similar results with …’ be ‘Our results similar to …’?
Reply 15: Thanks for very much. We have corrected this sentence as you suggested.
- Results and discussion (2.4): it is a bit strange in my opinion to mention Figure 4B before Figure 4A.
Reply 16: Thanks for your suggestion. We sincerely hope your understanding for this strange order, because we would better set the structure of Ulp1-Smt3 complex as Figure 4B for the most proper layout of the whole figure.
- Results and discussion (2.4): Line 247, should ‘Regrettably’ be ‘As expected’ since the di-glycine motif is highly conserved among eukaryotes and mutation of it should affect its maturation and cleavage?
Reply 17: We apologize for our negligence. We have corrected this sentence as you suggested.
- Figures 2B, 3B, 4A, 5B: some showed error bars, while some did not. What do the bars represent (SD, SE, or else)?
Reply 18: Thanks for your suggestion. Tiny errors in some figures are omitted because they presented nearly consistent cleavage efficiency. Error bars in figure 4A were added to show a larger span of data, representing SD of three independent experiments.
- Little to moderate change of English grammar and spelling check are required. For example, Line 38: their maturation requires à their maturation requires; Line 46: is benefit à is beneficial; Line 397, was download à was downloaded, etc.
Reply 19: Thank you very much for carefully reviewing our manuscript. We have sought help from English native speaker to carefully improve the writing of our manuscript. As a result, most of the sentences were revised and were highlight in the revised manuscript.

Round 2
Reviewer 1 Report
The authors have addressed my concerns (about a possible shift in the cleavage site upon mutating the following residue to glycine) by showing results from additional experiments and discussing them in the text.
Unfortunately, the new results are also generated by the same method as the old result, making them subject to the same potential problems. I would have preferred to see a more direct determination of the cleavage site by a Mass-spec based method. However, I would not make this suggestion a requirement.
I am also not totally happy with the language improvements. In particular the text inserted to document the newly added results could be improved for clarity. In particular, whas is meant by "Smt3P1’ P1 position" with the superscripted P1' and the non-superscripted P1 ?
Author Response
-Unfortunately, the new results are also generated by the same method as the old result, making them subject to the same potential problems. I would have preferred to see a more direct determination of the cleavage site by a Mass-spec based method. However, I would not make this suggestion a requirement.
Reply: Thanks for your professional comments. The purpose of characterizing the specificity of Ulp1 on Smt3 P1 and P2 when P1’-Ala was mutated into Gly is to elaborate that P1’ position Ala is essential to maintain the high specificity of Ulp1 towards Smt3-P1P2↓A. When this essential position was mutated to Gly, the possibly shifted and wobbling cleavage have verified with an assisted experiment. I think it is not a good business to spend more time and expensive cost to re-verify this ancillary experiment.
-I am also not totally happy with the language improvements. In particular the text inserted to document the newly added results could be improved for clarity. In particular, whas is meant by "Smt3P1’ P1 position" with the superscripted P1' and the non-superscripted P1
Reply: Thank you very much. We have further revised the language according to your suggestion. The refined part has been highlighted in the re-submitted manuscript (Line 293-358).
Reviewer 2 Report
The authors have satisfactory answered to all my questions/concerns. I have no further questions.
Author Response
Thank you very much for carefully reviewing our manuscript and providing us these professional comments. We have further revised the language according to your suggestion.